# Unraveling the Heterogeneous Mutational Signature of Spontaneously Developing Tumors in MLH1^−/−^ Mice

**DOI:** 10.3390/cancers11101485

**Published:** 2019-10-02

**Authors:** Yvonne Saara Gladbach, Leonie Wiegele, Mohamed Hamed, Anna-Marie Merkenschläger, Georg Fuellen, Christian Junghanss, Claudia Maletzki

**Affiliations:** 1Institute for Biostatistics and Informatics in Medicine and Ageing Research (IBIMA), Rostock University Medical Center, University of Rostock, 18057 Rostock, Germany; yvonne.gladbach@uni-rostock.de (Y.S.G.); mohamed.hamed@uni-rostock.de (M.H.); anna-marie.merkenschlager@uni-rostock.de (A.-M.M.); fuellen@med.uni-rostock.de (G.F.); 2Faculty of Biosciences, Heidelberg University, 69120 Heidelberg, Germany; 3Division of Applied Bioinformatics, German Cancer Research Center (DKFZ) and National Center for Tumor Diseases (NCT) Heidelberg, 69120 Heidelberg, Germany; 4Department of Internal Medicine, Medical Clinic III - Hematology, Oncology, Palliative Medicine, Rostock University Medical Center, University of Rostock, 18057 Rostock, Germany; leonie.wiegele@uni-rostock.de (L.W.); christian.junghanss@med.uni-rostock.de (C.J.)

**Keywords:** tumor mutational burden, exclusive/shared mutations, predicted tumor antigens

## Abstract

Mismatch repair deficient (MMR-D) tumors exemplify the prototypic hypermutator phenotype. Owing to the high mutation rates, plenty of neo-antigens are present on the tumor cells’ surface, ideally shared among different cancer types. The MLH1 knock out mouse represents a preclinical model that resembles features of the human MMR-D counterpart. While these mice develop neoplasias in a sequential twin-peaked manner (lymphomas > gastrointestinal tumors (GIT)) we aimed at identification of underlying molecular mechanisms. Using whole-genome sequencing, we focused on (I) shared and (II) mutually exclusive mutations and describe the process of ongoing mutational events in tumor-derived cell cultures. The landscape of MLH1^−/−^ tumors is heterogeneous with only a few shared mutations being detectable among different tumor entities (*ARID1A* and *IDH2*). With respect to coding microsatellite analysis of MMR-D-related target genes, partial overlap was detectable, yet recognizing shared antigens. The present study is the first reporting results of a comparison between spontaneously developing tumors in MMR-D driven tumorigenesis. Additionally to identifying *ARID1A* as potential causative mutation hotspot, this comprehensive characterization of the mutational landscape may be a good starting point to refine therapeutic concepts.

## 1. Introduction

Comprehensive genomic sequencing is an increasingly common practice in oncological precision medicine. The technological improvements in analyzing diseases at the genomic, transcriptomic, and epigenetic levels allow for identification of characteristic genetic changes. To detect disease-causing variants (so-called driver mutations) and discover therapeutic target genes, whole-exome sequencing (WES) is a highly meaningful sequencing method [1]. By applying this technique, the coding region of the genome is captured and sequenced at high depth. The interrogation of the exome in clinical diagnosis raises challenges of uncovering mutational profiles of heterogeneous tumors and may allow the development of new immunotherapeutic approaches [2,3].

Closely linked to hypermutation and high immunogenicity is the *MLH1* gene, belonging to the DNA mismatch-repair (MMR) family. Germline, as well as somatic *MLH1* mutations drive tumorigenesis. The hallmarks of the resulting tumors are mismatch-repair deficiency (MMR-D) and an exceedingly high tumor mutational burden (TMB) [4]. The latter is characterized by frameshift mutations and, in consequence, by functionally impaired proteins as well as elevated frequencies of non-synonymous mutations. A high neoantigen load reflects high TMB. The tumor cells’ MHC molecules, capable of eliciting anticancer T cell responses, present these mutated self-peptides. However, immune evasion, such as loss of the MHC class I subunit, beta-2-microglobulin as well as upregulation of immune-checkpoint molecules (PD-1/PD-L1) is quite common and most tumor-infiltrating T cells show signs of exhaustion [5].

Clinically, MMR-D is associated with sporadic as well as hereditary cancer, with the latter being due to mono- or biallelic MMR germline mutations. Individuals with monoallelic germline mutations suffer from Lynch syndrome and have an 80% lifetime risk of developing cancer [6]. The bi-allelic counterpart is constitutional mismatch-repair deficiency (CMMR-D), associated with a complex and slightly different tumor spectrum than that seen in Lynch-associated cancers [7,8]. As there is a high likelihood of tumorigenesis for both syndromes, attempts to delay or even prevent tumor formation are ongoing (e.g., *clinicaltrials.gov.identifiers*: NCT01885702; NCT02813824; NCT02497820; NCT03070574). Moreover, preclinical tumor models may provide new insights into tumor biology, helping to refine prevention and therapy.

MLH1^−/−^ knockout mice constitute an ideal preclinical model for hereditary tumor syndromes. With a virtually 100 % penetrance, these mice develop a variety of malignancies, including highly aggressive lymphomas in different organ sites, gastrointestinal tumors, and malignant skin lesions [9,10]. The sequential appearance of these neoplasias, often in a mutually exclusive fashion (“either-or” principle in terms of tumor type) raises the question of how the biological and/or molecular mechanisms of these neoplasias relate to each other. To gain insights into the underlying molecular alterations that affect tumor formation and thus act as drivers, we herein analyzed the number and type of somatic mutations arising at the bottom of a germline MMR-D. Exclusive as well as shared alterations were identified, explaining the heterogeneous clinical presentation.

## 2. Results

### 2.1. In Vivo and Ex Vivo Data

MLH1^−/−^ mice are prone to different tumors. Data from 90 mice reveal a heterogeneous distribution pattern and the median age of onset varies, with a high prevalence of early Non-Hodgkin T cell lymphomas, primarily arising in the thymus and spleen (43.5 and 18.5%, respectively), after which mice develop epithelial tumors of the gastrointestinal tract (GIT) tumours, primarily located in the duodenum or lymphoid skin lesions (26.0 and 12.0%, respectively). Besides, gender-specific differences are apparent with female mice developing lymphomas more frequently than GIT and vice versa in males (Figure 1A).

For our comprehensive analysis, we included a set of four tumor samples, mainly covering the spectrum observed in MLH1^−/−^ mice (Table 1). Since all MLH1^−/−^-associated tumors harbor MHC class I on their surface, which is preserved in the resulting in vitro cell culture as well as in the allograft [9], phenotyping focused on immune evasion markers, such as PD-L1, CTLA-4, LAG-3, and TIM-3. As anticipated, lymphomas expressed all markers in high abundance (Figure 1B). By contrast, PD-L1 and CTLA-4 surface expression on GIT was heterogeneous and found on only 20-40% of cells, respectively (Figure 1B; referring to P11 and P12, respectively). LAG-3 and TIM-3 expression was confined to tumor-infiltrating lymphocytes and thus below 5% (Figure 1B; referring to P14 and P15, respectively and [3]). Generally, lymphocytic infiltrates in MLH1^−/−^-associated GIT showed an equal distribution between CD3^+^CD4^+^ and CD3^+^CD8^+^ T cells (Figure 1C, right bars with dashed lines and representative immunofluorescence images). However, marked differences were found between the two GIT cases #7450 and #328, nicely reflecting the heterogeneity even among GIT with similar genetic background. As can be taken from Figure 1C, the value of both cell types in #7450 (dotted plot) exceeds the mean value, while in #328 (checkered plot) infiltrating cell numbers are far lower than on average.

### 2.2. Mutational Landscape of MLH1^−/−^ Tumors

The different tumor spectra detectable in MLH1^−/−^ mice prompted us to perform a whole-exome sequencing analysis. This experiment confirmed our observations of high TMB. For the confirmatory analysis, we initially considered all point mutations. TMB ranged from 39 mutations/Megabase (MB) (in GIT #7450) to 943 mutations/MB (in GIT #328). Breaking this down to non-synonymous substitutions, i.e., missense and nonsense mutations, high mutational burden was, by the definition of ref [11] (>10 mut/MB), preserved, with 3/4 samples even being ultra-hypermutated (>100 mut/MB) (Figure 2A).

Most mutations in cancer genomes, however, are ‘passengers’ and do not bear strong imprints of selection. Apart from this, the status of hypermutation complicates the identification of driver mutations simply due to the sheer abundance of somatic variants [12]. As MLH1^−/−^-associated lymphomas showed a more aggressive in vivo growth behavior and develop considerably earlier than GIT, we expected a different molecular make-up of the lymphomas and possibly also a higher mutational burden. While the former was actually true, the mutational load was not generally higher. Indeed, the lymphomas’ mutation rates were highest, but the two GIT cases #7450 and #328 itself differed significantly from each other (Figure 2). The #328s’ mutational load was mostly comparable with that of the splenic and skin lymphomas (Figure 2). The very early formation of this GIT before the mean age of onset (~33 weeks for GIT), may explain our finding best (Figure 2B). Also, there was an inverse correlation between TMB and age of onset (Figure 2B).

Missense mutations were more frequent than nonsense mutations, and base changes were mainly due to transitions (C > T; A > G) (Figure 2A). High levels of transversion are associated with PD-L1 abundance that functions as a biomarker for immune-checkpoint inhibition. In a previous study, PD-L1 abundance was analyzed on MLH1^−/−^-tumors [13]. In line with our recent findings on high percentage numbers of transitions instead of transversions, we also detected only low-moderate PD-L1 expression (<30%), and if detectable, this was mainly confined to infiltrating immune cells, see ref. [13].

### 2.3. Mutation Types and Shared Mutations

Comparative whole-exome sequencing of the MLH1^−/−^ GIT #7450 and #328 with the lymphoma revealed the mutanome of these samples (Figure 3 and Figure 4). Considering the number of mutations per type, marked differences were found in the #328 compared to the #7450 GIT and both lymphomas (Figure 3A). As anticipated, the number of silent mutations was high whereas the nonsense mutation rate was low in all four tumor samples. Focusing on the non-synonymous mutations demonstrated similar amounts of these for the #328 GIT and the splenic lymphoma, but slightly increased numbers compared to the skin lymphoma. In fact, #328 from a male mouse had the highest mutation rate in all mutation types, compared to the female counterpart GIT and to the other cancers. Furthermore, lymphomas and GIT shared 640 mutations (Figure 3B), although a high amount of single nucleotide variations (SNV) for each case was exclusive. Interestingly, the lymphomas shared a higher amount of SNV with 684 vs. only 10 in the GIT.

In dissecting the mutational landscape (Figure 3C) we initially focused on shared mutations among all cancers most likely functioning as drivers for MLH1-associated tumorigenesis. With this analysis, a surprisingly low number of shared mutations were detectable. These included mutations in the following genes: *AT-rich interaction domain 1A (ARID1A)* and *isocitrate dehydrogenase 2* (*IDH2). ARID1A*, located on chromosome 4, encodes a protein of the SWI/SNF family, contributing to the large ATP-dependent chromatin-remodeling complex. It is an epigenetic modifier that functions as a tumor suppressor; gene mutations have been reported in several malignancies [14]. *IDH2* is different, usually being amplified and/or overexpressed in tumors. Mutations are associated with poor clinical outcome, well reflecting the clinical presentation of biallelic MMR-D-driven malignancies.

### 2.4. Mutational Frequencies and Types of Alterations for Selected Gene Sets Based on Prior Knowledge in Mouse and Human

Next, we focused on selected genes with known relevance for tumor initiation and progression as well as tumor-suppressive function (Figure 4A). In this analysis and in accordance with the human counterpart, MLH1^−/−^-associated tumors harbor additional mutations in *PIK3CA*, *BRAF* and/or *KRAS* and *ERBB3* [15]. Therefore, we created an oncoprint of the non-synonymous alterations in this selected gene set (Figure 4A). A missense mutation was found for *PIK3CA*, *BRAF*, *KRAS*, and *ERBB3*. Events associated with these genes show alterations distributed in a nearly mutually exclusive way across all samples. Unexpected are the alterations in the *NF1* gene, a negative regulator of the *RAS* signal transduction pathway that is modified in #328 (primary and cell line). These *NF1* mutations were shown to be concomitant with mutations in the oncogenes *BRAF* or *KRAS* [16,17]. Here, the same samples that had *NF1* mutations displayed *BRAF* mutations as well, but only the skin lymphoma and the GIT (#328) had additional *KRAS* mutations. In the case of *APC*, a negative regulator of the *Wnt* signaling pathway [18], we observed acquired additional missense mutations in the #7450-derived cell line that had not been detected in the primary.

The *PI3K* pathway is affected by *PIK3CA*. Further interactions of *PIK3CA* are with *AKT* and *mTOR* pathways, mutated in one GIT (#328) and both lymphomas. In gastric cancer, *PIK3CA* mutations are associated with a higher aggressiveness [19], while for the lymphomas a strong correlation with *PTEN* mutations is described [20]. Both genes were found to be mutated in the GIT (#328) and in splenic lymphoma, respectively. The same was also true for *EGFR* signaling members *ERBB2* and *ERBB3*.

*SMAD4* and *POLE* were exclusively detectable in lymphomas (Figure 4A). Of note, the detected somatic *POLE* mutation was located within the exonuclease domain of this gene, executing a proofreading function to decrease the mutation rate during DNA replication. In conjunction with germline MMR-D, *POLE* mutations strongly increase the number of gene mutations in affected tumor cells. Additionally, *SMAD4* acts as a tumor suppressor and inhibits transforming growth factor-β-mediated signaling. Mutations are associated with poor outcome and high metastatic potential. With respect to MLH1^−/−^ tumors, *SMAD4* mutations were constrained to lymphomas.

### 2.5. Pairwise Exclusively SNV in GIT and Lymphomas

The above data identify different and common mutations among MLH1^−/−^ cancers. We examined the pairwise exclusive mutations among GIT or lymphomas in-depth (Figure 3C). While the #328 GIT had fewer mutations per gene, this tumor had generally more genes affected such as *APC*, *PTEN*, *ERBB4*, *IDH1*, *PIK3CA*, *MET*, *BRAF*, *KRAS*, *ERBB3*, *NF1*, *CDC27*, *SOX9*, *MSH3*, *MIER3*, and *RBFOX*. Remarkable is that although both primary tumors have shared genes affected by mutations, the positions of the SNV are exclusive for each of the tumors (shown in red with the corresponding annotation containing the position and the gene, all SNV are in yellow), likely implicating random distribution during oncogenesis.

A pairwise comparison of the splenic and skin lymphomas showed that they share the mutated genes *APC*, *SMAD4*, *PTEN*, *ERBB4*, *PIK3CA*, *ARID1A*, *BRAF*, *MET*, *IDH2*, *ERBB3*, *NF1*, and *CDC27*. However, exclusive changes were also observed. Each of the lymphomas has SNV in six exclusive genes. Therefore, we analyzed in detail the processes and functions the exclusively mutated genes are involved in. The splenic lymphoma exhibits SNV in *MYO1B*, *CTNNBL1*, *ERBB2*, *SOX9*, *MSH3*, and *RBFOX1*, involved in cell migration, the spliceosome, cellular responses, chondrocytes differentiation, and MMR. Exclusive SNV for the skin lymphoma were found in the following genes: *IDH1*, *CASP8*, *ACVR2A*, *KRAS*, *IGF2*, and *POLD1*. All of these genes are part of the electron acceptor, cell apoptosis, oncogene, growth, replication, and repair.

### 2.6. Prevalence and Hotspot Regions of GIT and Lymphomas in ARID1A, POLE, and SMAD4

Mutational hotspots provide more insights into the underlying biology together with the mutations in the domains, such as binding domains. Mutations of major interest for the GIT and lymphomas are appearing in *ARID1A* (Figure 4B). Significant differences between the GIT and lymphomas are the loci of the mutations as well as that more loci are affected in the lymphomas. One amino acid change, D2046N, is located in the SWI/SNF-like complex subunit BAF250/Osa, the ATP-dependent chromatin remodeling complex that regulates gene expression through nucleosome remodeling. Furthermore, the splenic and skin lymphomas share the amino acid changes S571P and P585S. All other mutated loci are found in the putative disordered regions of the *ARID1A* gene.

Figure 4B illustrates the mutational hotspots of the mutually exclusive lymphoma-associated genes *POLE* and *SMAD4*. In *POLE*, we found the shared amino acid change D262Y for the splenic and skin lymphoma. Furthermore, the splenic lymphoma has a nonsense mutation, E265*, in the DNA polymerase B exonuclease domain that adopts a ribonuclease H type fold [21]. Interestingly, the lymphomas show an amino acid change, V346I, in the MH2 domain implicated in the control of cell growth [22] of *SMAD4*. Additional loci are in the putative disordered regions, T211A and A229G.

### 2.7. Somatic Mutation Distribution and Nucleotide Changes in Chromosome X

In addition to the TMB calculations, we analyzed the mutational load per chromosome. This mutational load was normalized based on the size of the respective chromosome, and the mutation rate per MB plotted for all four entities (Appendix A). Therefore, this presentation of the chromosomal distribution of SNV shows that for all entities, chromosome 11 accumulated the highest numbers of somatic mutations per MB. Given the fact of differences in gender and a possible hyper-mutation of chromosome X, we investigated chromosome X further for the nucleotide changes in pyrimidines C and T (Appendix A). C is also responsible for the deamination of the X-chromosome. The coloring of the bars represent the proportions of the six possible nucleotide changes of the pyrimidines (C > A, C > G, C > T, T > A, T > C, and T > G) for chromosome X. Comparing the GIT, we see that T > C is less frequent in the #328 than in the #7450. In turn, the changes C > A and T > G are more frequent. The lymphomas show only minor differences in the nucleotide changes for C > T and T > C, whereby the skin lymphoma had slightly lower changes in this latter than the splenic lymphoma.

### 2.8. Microsatellite Instability (MSI) Pathway—Prognostic, Predictive and Therapeutic Implications

To identify target genes for neoantigen-based vaccination approaches, we analyzed the SNVs in genes of the MSI pathway [23,24] as well as genes associated with the MSI status [25] and survival (based on hazard ratio in the human counterpart) (Table 2 and Table 3). The SNVs shared among all cancer entities were found in *ARID1A, BCL2, PIK3CA*, and *SMAD4*. ARID1A shows more silent than missense SNVs for the 7450-GIT, whereby all the other show missense SNVs. These genes are associated with overall survival (OS), disease-free survival (DFS), and disease-specific survival (DSS), having SNVs, the prognosis gets worse. The same holds true for PIK3CA, but it is only associated with recurrence-free survival. SMAD4, however, ensures a slightly better prognosis, since it endeavors only silent SNVs. BCL2 shows more clearly the differences between the female and male mice since it has SNVs that are not classifiable, whereby the other show silent with or without additional missense SNVs. It would ensure at least an event-free survival.

Poorer prognosis is expected for the lymphomas, due to the identification of exclusive genes within the MSI pathway, MSI status and survival associated genes. The same SNV classes have been identified for *AKT2, EGR1, INADL*, and *MAPK1* lowering a good prognosis for survival due to their association with OS, RFS, and EFS. Besides come same SNV classes in *CASP4* (OS), *CERS1* (RFS), and *MAPK1* (OS and RFS). Remarkable are the different SNV classes for *CASP9* and *ECHS1*, not associated with survival yet. For *CASP9*, there is a change from a missense SNV to a not classified impact, and the same goes for *ECHS1*, whereby the change is from silent SNVs to a not classified impact.

### 2.9. Ongoing Mutations in MLH1^−/−^ Tumors

To pursue the ongoing event of acquiring novel mutations directly, we subsequently compared the mutational profile of selected primary tumors and their corresponding cell line upon in vitro culture. As anticipated and in conjunction with observations from clinical CMMR-D samples, numbers of somatic mutations continually increased (Figure 5A). As for the GIT case #7450, which harbors the lowest mutation numbers, TMB more than doubled in the derived cell line. The same was true for coding microsatellite (cMS) mutations, showing a gradual increase during in vitro passage (Figure 5B).

Finally, the cMS mutational profile was examined on a panel of primary tumors, i.e., lymphomas and GIT (*n* = 10 cases each). With this analysis, mutations were detectable in 15/26 markers, with 5 markers being shared among both cancer types (Figure 5C). Of note and of particular relevance, only two genes had the same mutation in their cMS marker of an MSI target gene, thus being in conjunction with our WES data of minor entity-overlapping mutational events. These are *AKT3*, a RAC-gamma serine/threonine-protein kinase with relevance in the maintenance of cellular homeostasis (balancing survival and apoptosis) and the endonuclease *ERCC5* (Excision Repair 5), involved in DNA excision repair. Mutations in this gene increase susceptibility for skin cancer development. Given the biological function of these two genes, their applicability as a target structure is an issue worth being determined.

### 2.10. In Vivo Vaccination Approach

Finally, mice with established GIT were included in a vaccination trial. Based on our previous study on minor entity-overlapping therapeutic potential between GIT and lymphomas [3], we here focused on cellular lysates made of two different GIT cell lines, namely MLH1^−/−^ A7450 and 328.

Repetitive application of the lysate prolonged survival of both treatment groups (Figure 6). Direct comparison of the individual lysates revealed better outcome in the MLH1^−/−^ A7450 treatment group. Median survival was 8.3 weeks (vs. 4.0 in the control group) and thus statistically significant longer even compared to the 328-treatment arm. Although not analyzed in detail here, accompanying in vivo imaging at day 28 of therapy revealed disease control only in MLH1^−/−^ A7450 vaccinated mice, while tumors in 328 treatment arm tended to progress (*personal observation*).

These findings confirm our molecular data on a biological level, in which the mutational signature predicts vaccination efficacy.

## 3. Discussion

Due to the biallelic germline MMR-D and the accordingly complex tumor spectrum, MLH1^−/−^ mice are frequently considered a better model for CMMR-D, instead of Lynch Syndrome. In men, *MLH1* has the highest frequency among monoallelic MMR mutation carriers and also the highest cumulative risk to develop any cancer [26]. Despite some differences, MLH1^−/−^ mice combine several key features of both syndromes. The recently detected heterogeneity of Lynch-associated cancers, being highly immunogenic or not (i.e., “hot”, “warm” or “cold” depending on the mutational load and signature) [27], additionally argues in favor of using these mice as a model for Lynch Syndrome as well. All malignant human CMMR-D cancers are ultra-hypermutated. However, this is so far not known for tumors arising in the context of Lynch syndrome. Apart from this, there is growing evidence of individual differences between patients, likely being dependent on the underlying MMR defect [28]. In line with this broad heterogeneity among MMR-D-related malignancies, we here show that MLH1^−/−^ tumors (I) are either hyper- or ultra-hypermutated, (II) have an entirely individual mutational profile and (III) harbor different mutational hotspots in affected genes.

Of note, along with the different sites of manifestation, a gender-specific prevalence is evident in mice, where thymic lymphomas primarily develop in females, and 80% of GIT manifest in males. Apart from hormonal-differences (i.e., estrogen-pathway activation vs. androgen-pathway activation), X-chromosome specific alterations may provide a rationale for the different prevalence. The X-chromosome of MLH1^−/−^ mice is indeed highly mutated. However, we could not find evidence for a hypermutated X-chromosome. We hypothesize that the mutation rate as well as the type of mutations on the X chromosome is a cancer-specific feature for the lymphomas and represents a subgroup in the GIT primary tumors. This somatic mosaicism is the result of “self-promoting” mutations that favors (or hinders) expansion of the mutant clone and shapes their resulting genomic make-up in males or females. In addition to the gender, several biological characteristics argue in favor of this hypothesis, leaving technical artifacts as the sole or dominant explanation for the different mutational load very unlikely: (I) the age of onset; (II) the number of tumor-infiltrating lymphocytes in the primary and (III) the experimental follow-up study showing different vaccination efficacy in tumor-bearing MLH1^−/−^ mice. In here, the GIT lysate from the #7450 case was much more effective than the tumor lysate of #328. Hence, the mutational signature likely predicts vaccination efficacy. Still and without any doubt, this has to be confirmed in a larger series on more samples per entity and gender.

In the only prior study on the mutational landscapes of MLH1^−/−^ murine tumors, spontaneous as well as irradiation-induced T cell lymphomas were analyzed. Results identified many genes commonly mutated but not previously implicated, such as genes involved in NOTCH (e.g., *Notch1*) and PI3K/AKT (e.g., *Pten, Akt2*) signaling pathways as well as frameshift mutations in mononucleotide repeat sequences [10]. Our in-depth sequencing analyses adds to these findings and broadens the knowledge on the mutational spectrum of MMR-D related malignancies apart from lymphomas.

By performing in-depth sequencing on lymphomas and GIT, most altered genes involved *APC*, *PTEN*, *PIK3CA*, *Her2*-related genes, *BRAF* and/or *KRAS*. However, it is often difficult to determine, which mutations arose first or whether their order is essential in driving tumorigenesis. While MLH1^−/−^ mice are inbred and all descendants harbor the same germline mutation, we expected a comparable mutational profile among cancers. This was, nevertheless, not the case with a comparably low number of shared mutations among analyzed cancer types. Indeed, entity-overlapping mutations were exclusively detectable in *ARID1A* and *IDH2*. In men, *IDH2* mutations are linked to brain tumors and acute myeloid leukemia, while *IDH1* mutations may drive tumor progression. IDH-mutated tumors are clinically and genetically distinct from their wildtype counterpart. *ARID1A* is a bona fide tumor suppressor and a frequent mutational hotspot in many cancers, including those of the breast, gastrointestinal tract and ovary [29]. *ARID1A* functions in MMR regulation, as its loss is associated with MSI. It is supposed to be a driver gene, occurring secondary to MMR-D in gastric cancers, representing an alternative oncogenic pathway to p53 alteration [30]. Neither of the tumors examined here showed *TP53* mutations, confirming this assumption. *ARID1A* mutations in MLH1^−/−^-associated tumors are distributed along the entire length of the gene, thus distinguishing it from typical MSI target genes, a finding also described in the human counterpart [31].

Loss of *ARID1A* shortens time to cancer-specific mortality in men, where mutations are associated with both sporadic and hereditary MSI cancers [32]. Though it is still unknown whether CMMR-D cases harbor *ARID1A* mutations as well, this observation fits with our finding on MLH1^−/−^ mice having a significantly shortened life expectancy.

Comparable analysis done by others on human cancers revealed ten overlapping mutations, all of which are InDels involving G7 or C6 repeats [33]. These mutations are potentially caused by MSI, and although not analyzed in detail here, we also observed *ARID1A* mutations in our own collection of MMR-D-associated clinical CRC cases (22% of cases harbor mutations in the G7 repeat and have additionally wildtype *TP53*). *ARID1A* is thus a likely target for personalized medicine in cancer treatment. *ARID1A*-deficiency-based synthetic lethal, targeted drugs are under clinical investigation to suppress cell growth and promote apoptosis [33] (see also *clinicaltrials.gov*). Whether this provides a real opportunity for *ARID1A*-deficient MMR-D-related cancer will need to be tested prospectively.

Another interesting finding of this study was the high prevalence of transitions instead of transversions among gene mutations. Although this is commonly seen in non-smoking associated cancers, we wish to stress the close link between transitions (mainly C-to-T) and MSI as well as *POLE/POLD1* mutations in men [34]. While these alterations are detectable in MLH1^−/−^-associated tumors similar mechanisms can be assumed. As for the latter, the relevance of somatic *POLE/POLD1* mutations in the context of MMR-D has only begun to be enlightened. In clinical cases of CMMR-D, most tumors harbor *POLE* mutations affecting the exonuclease domain or domains important to the intrinsic proofreading activity [35]. These cells rapidly and catastrophically accumulate point mutations, yielding the ultra-hypermutated phenotype – quite similar to our observations. Cells mutate continuously, raising the question of how many mutations can be tolerated until a maximum threshold is reached that creates the Achilles’ heel.

All mutations described so far may have prognostic, predictive and surveillance potential. Besides, a plethora of low-frequency variants whose functional consequences and clinical actionability are unknown were found as well. We thus conclude that for MLH1^−/−^-driven oncogenesis, many mutations are mere passengers with no functional consequence.

A point worth mentioning is the fact that MMR-D associated lymphomas in men are rarely documented in the literature and most of them develop in patients suffering from CMMR-D. In here, however, hematological malignancies, are common and even in this small group of patients, mostly associated with founder *MLH1* and/or *MSH2* mutations [36]. A recent study described the occurrence of MSI/MMR-D in de novo Hodgkin lymphoma, but at a significantly higher frequency in Hodgkin’s lymphoma survivors developing therapy-related colorectal cancers [37]. *MLH1* promoter hypermethylation, but also double somatic mutations are the leading causes of MMR-D [38]. Highlighting the relevance of *MLH1,* one study also demonstrates the failure of a specific immune-chemotherapy regimen in follicular lymphoma patients showing *MLH1* gene polymorphisms [39]. In addition to germline MMR-D, the frequency of MMR-D arising secondary to anticancer treatment, is thus supposedly higher than previously predicted, considering this a clinically relevant finding. This may finally witness a resurgence of interest to apply and develop alternative therapeutic regimens.

With respect to our study, some neoantigens are auspicious. These include AKT3, ERCC5 and maybe ARID1A as general MMR-D targets, but also POLE as lymphoma-specific neoantigen. The finding that this particular mutation was confined to lymphomas may not only explain why this type of malignancies arises earlier in mice than GIT but also represent a target structure worth being included.

Summarizing our findings, we identified a gender-specific aspect in the tumor spectrum, with resulting tumors displaying their own idiosyncratic mutational landscape. The identified shared and mutually exclusive mutations warrant further investigations for therapeutic approaches ideally combined with immune-stimulating and/or -restoring agents.

## 4. Critical Limitation of the Study

A major limitation of the current study is the low sample number of each tumor type included is WES analysis (*n* = 1 plus 2 sample-derived cell lines at very low passage). Nevertheless, this is a pilot study aiming to explore the mutational signature of MLH1^−/−^-associated murine tumors - comparable analyses are to the best of our knowledge not documented in the literature. With respect to cMS mutations, which were done on a larger cohort, quite similar heterogenic distribution was found between individual tumor samples. Another limitation is the fact that all mice are inbred and thus harbor the very same germline mutation in the *MLH1* gene. In men, the germline mutational spectrum is much more complex (i.e., *MLH1, MSH2 +/− EpCAM, PMS2*, and *MSH6*) and heterogeneous (different hotspots, with some of them still being classified as variance of uncertain significance). As a result, patients present with a diverse clinical presentation/organ manifestation and age of onset. Therefore, a direct comparison with human samples is hardly realizable. Still, the heterogeneity observed between individual MLH1-associated cancers well reflects the human situation and underlines the individual character of malignancies. By identifying some overlap between the murine and human oncoprint these analyses will likely help to get a deeper understanding of the genetic or genomic make-up of MMR-D-related cancers.

## 5. Materials and Methods

### 5.1. In Vivo Mouse Model and Sample Acquisition

Homozygous MLH1-deficient mice were bred in the animal facilities (University of Rostock) under specified pathogen-free conditions. Trials were performed in accordance with the German legislation on protection of animals and the Guide for the Care and Use of Laboratory Animals (Institute of Laboratory Animal Resources, National Research Council; NIH Guide, vol.25, no.28, 1996; approval number: LALLF M-V/TSD/7221.3- 1.1-053/12 and 026/17, respectively). Tumor samples for sequencing analysis were obtained from four individual MLH1^−/−^ mice upon spontaneous development. Two mice suffered from a lymphoma (spleen and skin), and the remaining two mice had gastrointestinal tumors both located in the duodenum. All tumors were at an advanced stage at resection. Samples were snap frozen in liquid nitrogen and stored at −80 °C. Isolation of gDNA was done by GATC (samples: 7450 primary GIT; A7450 P15 cell line) and CEGAT (samples: 328 primary; 328 cell line; 1351 lymphoma (spleen); 1444 lymphoma (skin)), respectively, prior to the sequencing analyses.

### 5.2. Whole-Exome Sequencing (WES) Analysis

Data preprocessing: First, the read quality of the raw sequencing data was investigated with FastQC v.0.11.5 [3] to adjust the adapter trimming. The sequencing adapters were trimmed using Skewer v.0.2.2 [40], for the reads after demultiplexing with Illumina bcl2fastq v.2.1919 (Illumina, San Diego, CA, USA). Reads with a Phred quality score ≥ 30 were considered as high quality. The whole-exome reads were mapped to the published mouse genome build ENSEMBL mm10.75 reference using Burroughs Wheeler Aligner (BWA) [41] with the bwa-mem algorithm v.0.7.2 [42]. Reads with multiple mappings and the same mapping score have been discarded from further analysis. Additionally, PCR duplicate reads have been removed by using Samtools v.0.1.18 [41] to prevent artificial coverage brought on by the PCR amplification step during library preparation and to avoid its impacts on identifying false positive mutations. SNV calling was performed using the Varscan v.2.4.2 [43], and the detected variants were annotated based on their gene context using snpEff v.4.3 [44]. To compare these data with data from our previously published paper [3], the mm9 SNV coordinates were converted to the corresponding mm10 coordinates with hgliftover v.30-10-2018 [45,46]. The cell line A7450 from the batch was processed likewise [3].

### 5.3. Data Visualization

For the MLH1^−/−^ GIT primary tumor, derived cell line, spleen and skin lymphoma, the summarized mutational profiles with the corresponding mutation type were summarized as stacked bar plot with ggplot2 [47], as Venn diagram with VennDiagram [48] as well as whole-genome ideograms with the circlize [49] R package. Based on the mm10 mouse assembly, provided by the circlize package [49], the filtered mutational profiles are presented as ideograms.

All four different mutational profiles were filtered for the exclusive SNV for each condition separately. Additionally, mutation filters such as the mutation type (missense and nonsense) as well as those mutations occurring in known annotated genes have been applied to mutational profiles.

Additional details like TSTV (transition and transversion) ratio, allele frequency spectrum, base changes, and variant quality for each mutational profile were obtained using the vcf.iobio tool [50].

Furthermore, mapping the mutations and its statistics on a linear gene product (proteins of interest) was done with a ‘lollipop’ mutation diagram generator [51]. Based on the knowledge from the human MMR-D counterpart and general involvement in tumorigenesis, the genes for further analysis were chosen with high probability to mutate.

### 5.4. Hypermutation and TMB Calculation

We adopted the following definition for calculating the TMB [52,53,54]:(1)$$\frac{\#\;all\;non-synonymous\;SNVs\;on\;target}{covered\;exome\;by\;sequencing\;kit}\;$$

Simply, we used intersectBed from the BEDTools suite v.2.26.0 [55] for ensuring the SNV being on target regions of the Agilent SureSelect Mouse All Exon Kit mm10 (Santa Clara, CA, USA) and then summing up the number of somatic mutations per MB of the transcriptome mm10 (48.3 MB).

We use the definition of hypermutation as follows [34,53,54]:(2)$$per\;MB_{cancer\;entity}=\left\{\begin{array}{c} hypermutated\\ ultra-hypermutated \end{array}\right.\begin{array}{c} \;if\;non-synonymous\;SNVs>10\\ \;if\;non-synonymous\;SNVs>100 \end{array}$$
where
(3)$$cancer\;entity\in \left\{GIT,\;derived\;cell\;line,\;lymphoma\right\}\;$$

### 5.5. Coding Microsatellite (cMS) Frameshift Mutation Analysis

Fragment length analysis was done from multiplexed PCRs of gDNA from tumor and normal tissue as described [9]. To identify potential MLH1 target genes, a panel (*n* = 26) was screened. Primers were designed using Primer3 software to yield short amplicons (≤200 bp).

### 5.6. In Vivo Vaccination with MLH1^−/−^ Tumor Lysates

Tumor cell lines MLH1^−/−^ A7450 and 328 were cultured and harvested prior to lysis using repetitive freeze/thaw cycles (*n* = 4). Protein lysates were gamma irradiated (60 Gy) and frozen immediately in aliquots at −80 °C before in vivo application. Thereafter, mice with gastrointestinal tumors (GIT) confirmed were conducted to vaccination according to a protocol described before [3]. Briefly, mice received four weekly injections of the vaccine (10 mg/kg bw, s.c.) in the first phase. Vaccination was continued (2.5 mg/kg bw, biweekly) until tumors progressed. Control mice were left untreated.

## 6. Conclusions

This study describes the mutational spectrum of MLH1^−/−^-associated tumors, spontaneously developing in mice. While these tumors arise at the bottom of the same germline mutation, the clinical presentations as well as resulting molecular alterations are heterogeneous, and, thus, likely being directly linked. By performing in-depth whole-exome sequencing analysis, we here identified, for the first time, a common mutational hotspot. *ARID1A* constitutes a potentially causative mutation, shared among different MLH1^−/−^-associated tumors and, thus, irrespective of the origin. This finding is of particular relevance for subsequent therapeutic and—even more important—prophylactic vaccination approaches aiming at entity-overlapping treatment of MLH1^−/−^-related tumors.

## Figures and Tables

**Figure 1 cancers-11-01485-f001:**
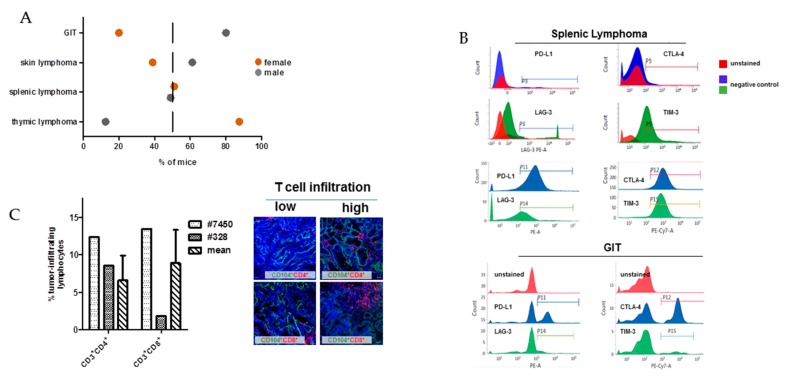
General distribution of tumors in MLH1^−/−^ mice and tumor phenotyping of immunoregulatory markers. (**A**) Type and frequency of tumors in MLH1^−/−^ mice; data were taken from a total of 90 mice in which spontaneous tumorigenesis was observed. Females were more likely to develop thymic lymphomas, while males were prone to gastrointestinal tract (GIT). Generalized splenic lymphomas were found in both genders. (**B**) Flow cytometric phenotyping of MLH1^−/−^ tumors reveals a high abundance of immune-checkpoint molecules on lymphomas. The phenotype of GIT is different with only a few cells expressing these exhaustion markers. Shown data refer to % numbers of cells measured in a live gate on total white blood cells. As controls, normal lymphocytes from healthy mice were stained with evasion markers (upper histogram plots) (**C**) Numbers of tumor-infiltrating lymphocytes in GIT as determined by flow cytometry. Left bar—#7450; middle bar - #328; right bar—mean value of tumor-infiltrating lymphocytes in GIT of different MLH1^−/−^ mice. Mean ± SD value result from *n* = 10 individual mice. Values are given as percentage of lymphocytes measured from 50,000 events in a live gate. Representative immunofluorescence images of GIT showing either low or high T-cell infiltration. Original magnification 20×.

**Figure 2 cancers-11-01485-f002:**
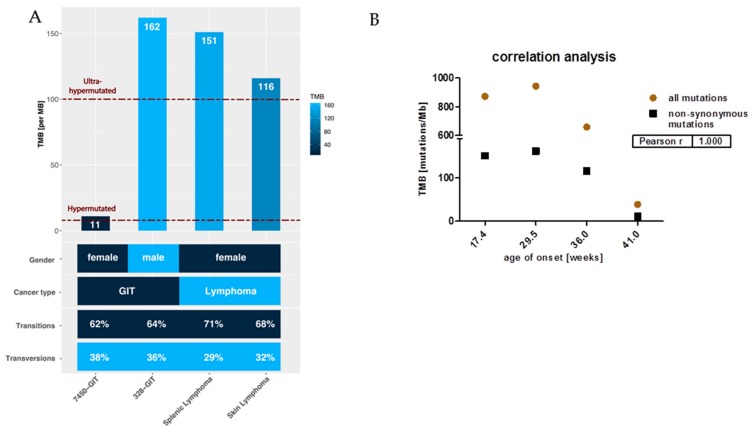
Mutational load and the ratio of non-synonymous mutations compared to all mutations in GIT versus lymphomas. (**A**) Each bar shows the tumor mutational burden (TMB) per Megabase (MB) for GIT and lymphomas, whereby cases of TMB > 100 are defined as ultra-hypermutated; these are GIT #328 and the lymphomas. Furthermore, in the heatmap, gender, cancer type, and the amount of transitions and transversions are shown in percent. (**B**) The number of mutations in relation to TMB and to the age of onset.

**Figure 3 cancers-11-01485-f003:**
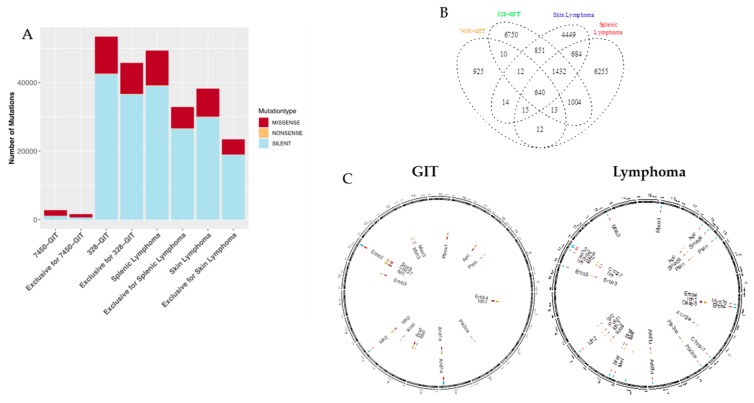
Mutational landscape and shared mutations in GIT versus lymphoma. (**A**) The statistics for each single nucleotide variations (SNV) type (missense, nonsense, and silent) are shown for the MLH1^−/−^ GIT primary tumors and the lymphomas including the distribution of the exclusivity of the SNV for each condition. (**B**) The GIT and lymphomas are showing 640 shared SNV in addition to their exclusive mutations. (**C**) Pairwise comparative mutational analysis of MLH1^−/−^ GIT and lymphomas. Ideogram plots show the genomic distributions of the missense mutations occurring in the annotated/known genes for the MLH1^−/−^ GIT (#7450 and #328, left side). The outermost track (track 0) is the cytoband of the mm10 assembly, followed by four tracks that visualize the SNV belonging to a specific condition (MLH1^−/−^ GIT primary tumors (#7450 and #328, spleen and skin lymphoma). All missense SNV for MLH1^−/−^ GIT primary tumors are blue (#7450, track 1) and yellow (#328, track 3) respectively with their corresponding coordinate on the mouse reference genome mm10 cytoband. The exclusive missense SNV for the MLH1^−/−^ GIT primary tumors are red (tracks 2 and 4) and annotated with a corresponding ID containing the SNV position and the affected gene name. Analogously to for the GIT, the splenic lymphoma (blue) and the skin lymphoma (yellow) are compared for all missense SNV in a pairwise fashion (right side). Red are the exclusive SNV respectively.

**Figure 4 cancers-11-01485-f004:**
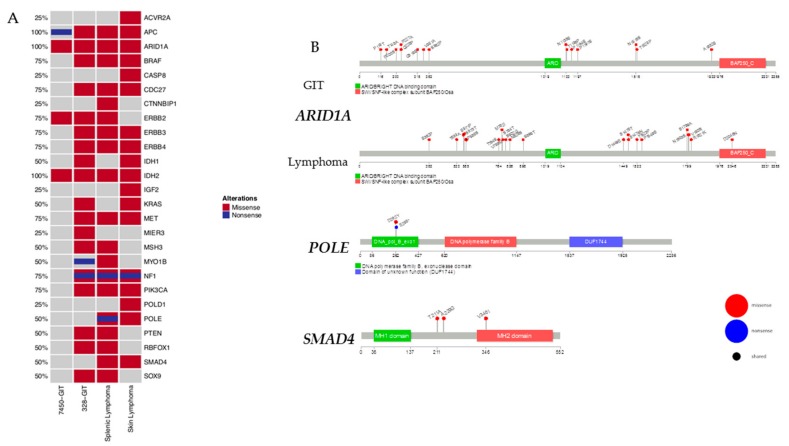
Specific cancer-related mutations in GIT and lymphoma. (**A**) The oncoprint presents genes with tumor-suppressive function as well as relevance for tumor initiation and progression. The analysis was done on primary GIT and lymphomas. This visualization provides an overview of the non-synonymous alterations in particular genes (rows) affecting particular individual samples (columns). Red bars indicate missense and blue bars nonsense mutations. (**B**) Prevalence and hotspot regions of GIT and lymphomas in *ARID1A*, *POLE*, and *SMAD4*. Known gene/protein domains are shown in color, other regions in dark grey. Non-synonymous mutations are depicted as nonsense by a blue lollipop and missense by a red lollipop, and a black circle in a lollipop indicates a shared locus of GIT and lymphomas entities. Each lollipop label shows the amino acid change and its location in the amino acid sequence.

**Figure 5 cancers-11-01485-f005:**
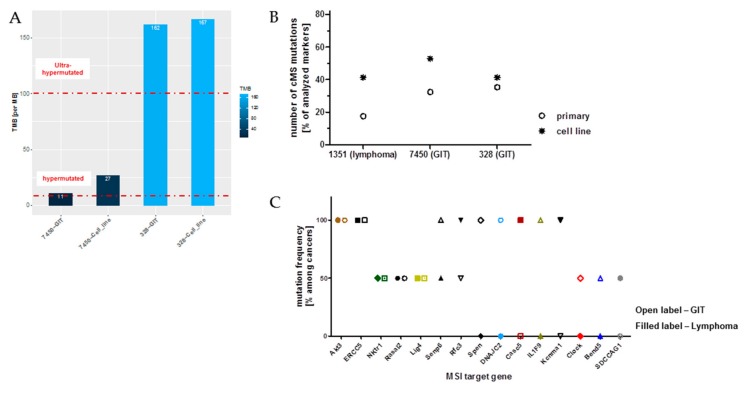
Mutational load and the ratio of non-synonymous mutations compared to all mutations in GIT versus the corresponding cell line. (**A**) The barplot shows the tumor mutational burden (TMB) per MB for the GIT and the corresponding cell lines, revealing ultra-hypermutation for GIT #328 and its corresponding cell line (TMB > 100). Furthermore, in the heatmap, gender, cancer type, as well as transitions and transversions are shown in percent. (**B**) Depicted are the percentages of cMS mutations in a panel of MSI target genes originally described in [9]. (**C**) Mutation frequency of individual cMS markers among GIT and lymphomas included in this study. Shared mutations were detected in 5/26 totally analyzed markers. Open label—GIT; filled label—lymphoma.

**Figure 6 cancers-11-01485-f006:**
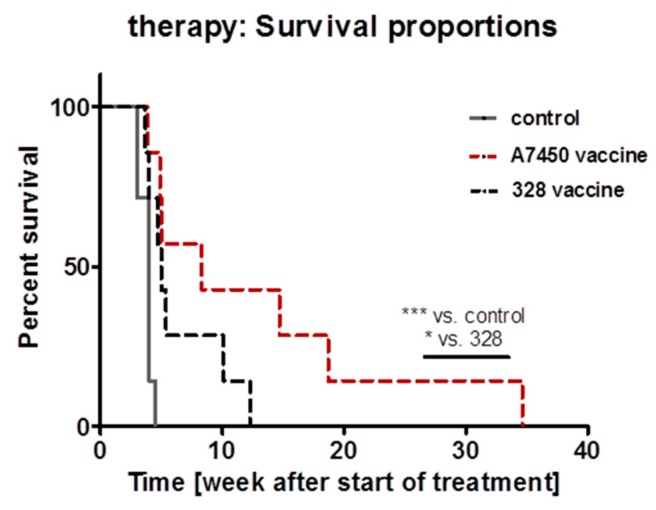
Kaplan–Meier survival curve analysis of vaccinated and control mice. Mice received repetitive local applications of the vaccine (10 mg/kg bw, s.c., *n* = 7 mice MLH1^−/−^ A7450 and *n* = 7 mice MLH1^−/−^ 328, respectively). Control mice were left untreated (*n* = 7 mice/group). *** *p* < 0.01 control vs. MLH1^−/−^ A7450 Log-rank (Mantel-Cox) Test; * *p* < 0.05 MLH1^−/−^ A7450 vs. MLH1^−/−^ 328 Gehan–Breslow–Wilcoxon Test.

**Table 1 cancers-11-01485-t001:** Tumor samples included in this study.

MLH1^−/−^ Number	Sex	Sample Type/Origin	Time of Onset [Weeks]
328	♂	GIT/duodenum	29.5
7450	♀	GIT/duodenum	41.0
1351	♀	Lymphoma/spleen	17.4
1444	♀	Lymphoma/skin	36.0

GIT—gastrointestinal tumor.

**Table 2 cancers-11-01485-t002:** Shared SNVs in MLH1^−/−^ cancer among the MSI pathway and potential association with survival.

Gene	7450-GIT	328-GIT	Splenic Lymphoma	Skin Lymphoma	Survival		
ARID1A	SILENT/MISSENSE	MISSENSE	MISSENSE	MISSENSE	OS	DFS	DSS
*BCL2*	NONE	SILENT/MISSENSE	SILENT	SILENT/MISSENSE			
*PIK3CA*	SILENT	SILENT/MISSENSE	SILENT/MISSENSE	SILENT/MISSENSE		DFS	
*SMAD4*	SILENT	SILENT	SILENT	SILENT/MISSENSE	OS		DSS

Based on COX *p*-value < 0.05; OS—overall survival; DFS—disease-free survival; DSS—disease-specific survival.

**Table 3 cancers-11-01485-t003:** Shared SNVs in MLH1^−/−^ lymphomas among the MSI pathway and potential association with survival.

Gene	Splenic Lymphoma	Skin Lymphoma	Survival		
*AKT2*	SILENT/MISSENSE	SILENT/MISSENSE	OS	DFS	DSS
*CASP4*	NA	NA			
*CASP9*	MISSENSE	NA			
*CERS1*	NA	NA		DFS	
*ECHS1*	SILENT	NA			
*EGR1*	SILENT	SILENT	OS	DFS	DSS
*INADL*	NA	NA			
*MAPK1*	SILENT	SILENT			
*RAC1*	SILENT	SILENT			DSS
*RHOA*	SILENT	SILENT			

Based on COX *p*-value < 0.05; OS—overall survival; DFS—disease-free survival; DSS—disease-specific survival.

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
