# Peer review of "Unraveling the Heterogeneous Mutational Signature of Spontaneously Developing Tumors in MLH1−/− Mice"

_cancers, 2019, doi:10.3390/cancers11101485_

Round 1
Reviewer 1 Report
In this manuscript, the authors investigated the mutational signature of different types of tumors in MLH1-/- mice, which mimic DNA mismatch repair deficient (MMR-D). With data from 90 mice, the authors reported that female MLH1-/- mice are more likely to develop thymic lymphomas while male mice are more prone to gastrointestinal tumors (GIT). Then they choose one male GIT, one female GIT, one female splenic lymphomas and one skin lymphomas sample for further study. By in-depth whole exome sequencing, the authors reported that different MLH1-/- tumors have distinct mutational profiles, even within shared mutated genes. They further analyzed the shared mutated genes and identified ARID1A as a potential tumor causative mutation. The idea is brilliant. This study is interesting and promising. However, the evidence is not strong enough to support the main conclusions. This manuscript can be improved in the following aspects.
Major concerns
A major problem of this manuscript is lack of sample numbers. In almost all the figures, there are astonishing differences between the male GIT sample #7450 with other samples. Can it be an outlier? It can be biased to draw a conclusion with only one sample in each group. It would also be interesting to compare mutational profiles between samples inside each group. Following the point above. The authors have not clarified what is the logic in choosing the samples. Are they randomly chosen or are they typical ones in each group? The rigor can be improved to choose at least two (three will be better) samples with medial age of onset and number of tumor-infiltrating lymphocytes for each group. The authors have highlighted the gender-specific differences though the manuscript. However, except Figure 1A which shows that female and male mice are more prone to develop different tumors, there is not much differences between male and female mice in other figures. The authors also have highlighted the X chromosome and claimed that the mutations/MB on the X-chromosome are almost twice compared to the mean mutation rate of the autosomes, which seems not the case according to supplementary Figure 1.Minor concerns
Figure 2B has not been referred in the text of the entire manuscript. Also, I don’t quite understand why the “age of onset” is embedded into other two charts as a secondary y-axis. If the reason is to show the correlation between TMB and “age of onset”, using the “age of onset” as x-axis may be better. The discussion about X-chromosome inheritance and inactivation between line307 and 312 is quite confusing. I don’t see any evidence of paternal-, maternal-, active-, or inactive X-chromosome from the results.Author Response
Dear reviewer,
Thank you very much for the excellent review of our manuscript "Unraveling the heterogeneous mutational signature of spontaneously developing tumors in MLH1-/- mice" by Gladbach et al. We appreciate the helpful comments and modified the manuscript according to the reviewer’s suggestions. Modifications in the text of the manuscript are marked by the track-changes option of Microsoft World.
In the following are our answers on a point to point basis:
Major concerns
The reviewer criticized the lack of sample numbers and the astonishing differences between the male GIT sample #7450 with other samples. She/he also asked if this might be an outlier.We thank the reviewer for this advice and completely agree that the number of samples is indeed a limitation of the current study. Nevertheless, we would like to stress the fact that this is a pilot study aiming to get an idea on the mutational landscape of different MLH1-/- tumors – comparable analyses are to the best of our knowledge not documented in literature. Unfortunately, such analyses are labor-intensive and thus hardly realizable in the given time-frame. Still, we made all efforts to unravel the differences found between the GIT samples and to provide a reasonable explanation. Therefore, we included a paragraph on genes affecting the Microsatellite instability (MSI) pathway and its prognostic, predictive and therapeutic implications based on recent publications in the human counterpart (page 8 and Table 2). We additionally incorporated some additional experimental data recently finalized in MLH1-/- mice. In this trial, MLH1-/- mice with established gastrointestinal tumors were vaccinated with the lysates of MLH1-/- A7450 (=GIT with hypermutated phenotype) or 328 (=GIT with ultra-hypermutated phenotype). Data have been included as Figure 6, showing a better survival upon MLH1-/- A7450 vaccination compared to 328 vaccinations (page 10). In line with our sequencing data and the biological results showing marked differences in (I) the age of onset and (II) the number of infiltrating lymphocytes in the original tumor, we here provide additional evidence that the mutational load and signature likely predicts vaccination efficacy. Also, these data nicely support our previous findings on only marginal entity-overlapping antitumoral capacity in the therapeutic situation (Maletzki et al., OncoImmunology 2017). Another fact worth mentioning is the only partial overlap in coding microsatellite mutations, which has been done on a larger sample number (N = 10 cases each; paragraph 2.9 in the enhanced version of the manuscript) here.
Still, we are aware that these data only partially answer the question and that another set of analyses is unquestionably a very interesting point that should be done; however, these additional analyses are out of the scope of this initial study. We integrated a section “critical limitation of the study” (pages 12/13), but apologize for not providing the information here. Nevertheless, we will be happy to incorporate data in an upcoming manuscript.
Following the same line. The reviewer asked for the logic in choosing the samples.
The rationale for choosing this set of samples is the fact that they are typical ones for MLH1-/- mice. As shown in figure 1A, mice develop a spectrum of lymphomas and GIT. We actively excluded thymic lymphomas, which are common as well because these were recently published in another study describing the mutational landscape of spontaneous and radiation-induced T-cell lymphomas (Carcinogenesis, 2019 Apr 29;40(2):216-224). When comparing results of this study, partial overlap was seen here, including PI3K/AKT (e.g. Pten, Akt2) signaling pathway alterations and frameshift mutations in mononucleotide repeat sequences within MSI target genes. Still, several differences are evident, underlining our results on an individual mutational signature even in the same genetic background.
Then, the reviewer criticized our statements on the gender-specific differences highlighting the mutational load on the X chromosome compared to the mean mutation rate of the autosomes.
We wish to thank the reviewer for this comment and the critical analysis of our results. With respect to the gender-specific differences, we highlighted this point because of the observation that female mice are much more prone to develop lymphomas than male mice. Though this is just an observation coming from our longstanding work with this particular mouse strain, we still considered this fact worth being mentioned in the manuscript. Also, a recent study described large sex-based differences in both TMB and T-cell inflammation of the tumor microenvironment in men (Conforti et al. Ann Oncol. 2019). In our study, we aimed at deciphering underlying molecular mechanisms apart from biological phenomena and started with the X- chromosome. Though the X-chromosome was highly mutated, we did not find evidence for a hypermutated X-chromosome as sole explanation. We thus conclude that the mutation rate as well as the type of mutations on the X chromosome is a cancer-specific feature for the lymphomas, representing a subgroup in the GIT primary tumors. This somatic mosaicism is the result of “self-promoting” mutations that favors (or hinders) expansion of the mutant clone and shapes their resulting genomic make-up in males or females.
Finally, we completely agree that our statement was a little overshooting and changed the discussion part accordingly (page 11).
Minor concerns
The reviewer noticed the missing reference of Figure 2B in the text.This has been changed accordingly (page 4). Also, a statement has been incorporated in the text: “Also, there was an inverse correlation between TMB and age of onset (Figure 2B).”
Then, the reviewer wished the “age of onset” as x-axis instead of embedding this fact into other two charts as a secondary y-axis.
This has been done accordingly (Figure 2B).
Another point raised by the reviewer is about some confusing discussion on X-chromosome inheritance and inactivation.
We thank the reviewer for this kind advice and removed this part from the main text (page 11).
Lastly, we were pleased to recheck english spell.
This was done throughout the manuscript.
We are very confident that the revised version now matches the requirements for publication in Cancers. We would be very pleased if you and the reviewers would find this enhanced version suitable for publication.

Reviewer 2 Report
In this manuscript, the authors present insightful meta-analysis data on mutational landscapes of spontaneously developing tumors in MLH1-/- mice. The information revealed in the mouse model could provide some insights into diagnosis and treatment of human cancers. Overall this work can be a good add-on to the field. However, I have the following comments and hope the authors could further strengthen the manuscript:
It’s very interesting to observe the gender differences in malignancies induced by MLH1 mutation. I hope the authors could provide more insights/explanations into this observation. For Figure 1B, I would suggest the authors to add a negative control which has low or no expression of immune evasion markers. This would help to confirm the cells are positive for these markers. Otherwise, it’ hard to tell if the staining is specific or not. In Figure 1C, mouse No. is too small for such analysis. I would suggest the authors to include more mice to confirm the results. Given the remarkable difference in mutational landscapes in female and male, as well as different tumors, I would suggest the authors to add extra mouse individuals for whole-exome sequencing. The authors should include at least one male and one female for each tumor type. If common mutations are observed in more individuals, this work will stand on a firmer ground.Author Response
Dear reviewer,
Thank you very much for the excellent review of our manuscript "Unraveling the heterogeneous mutational signature of spontaneously developing tumors in MLH1-/- mice" by Gladbach et al. We appreciate the helpful comments and modified the manuscript according to the reviewer’s suggestions. Modifications in the text of the manuscript are marked by the track-changes option of Microsoft World.
In the following are our answers on a point to point basis:
The reviewer wished to incorporate a negative control in Figure 1b.
We would like to thank the reviewer for this kind advice and included histogram plots of normal lymphocytes as negative control in Figure 1B (upper panel).
Then, the reviewer remarked the number of mouse for presenting % numbers of tumor-infiltrating lymphocytes.
Date presented in Figure 1 result from flow cytometric analysis of individual tumors #7450, #328, as well as an additional set of GIT resected from different MLH1-/- mice. For this latter data set, the mean + SD value is given which results from n=10 individual mice. Values are given as percentage of lymphocytes measured from 50.000 events in a live gate. With this analysis, we wanted to provide more evidence for the heterogeneity among these cancers. To get an impression, we incorporated representative immunofluorescence images of gastrointestinal tumors showing either low or high T-cell infiltration, which support our flow cytometric analysis (embedded in Figure 1C).
We changed the figure legend accordingly, but would like to keep these analyses as they are. If finally wished, these data can be removed from the manuscript.
Lastly, the reviewer wished to incorporate data on more mouse individuals for whole-exome sequencing.
We thank the reviewer for this advice and completely agree that the number of samples is indeed a limitation of the current study. Nevertheless, we would like to stress the fact that this is a pilot study aiming to get an idea on the mutational landscape of different MLH1-/- tumors – comparable analyses are to the best of our knowledge not documented in literature. Unfortunately, such analyses are labor-intensive and thus hardly realizable in the given time-frame. Still, we made all efforts to unravel the differences found between the GIT samples and to provide a reasonable explanation. Therefore, we included a paragraph on genes affecting the Microsatellite instability (MSI) pathway and its prognostic, predictive and therapeutic implications based on recent publications in the human counterpart (page 8 and Table 2). We additionally incorporated some additional experimental data recently finalized in MLH1-/- mice. In this trial, MLH1-/- mice with established gastrointestinal tumors were vaccinated with the lysates of MLH1-/- A7450 (=GIT with hypermutated phenotype) or 328 (=GIT with ultra-hypermutated phenotype). Data have been included as Figure 6, showing a better survival upon MLH1-/- A7450 vaccination compared to 328 vaccinations (page 10). In line with our sequencing data and the biological results showing marked differences in (I) the age of onset and (II) the number of infiltrating lymphocytes in the original tumor, we here provide additional evidence that the mutational load and signature likely predicts vaccination efficacy. Also, these data nicely support our previous findings on only marginal entity-overlapping antitumoral capacity in the therapeutic situation (Maletzki et al., OncoImmunology 2017). Another fact worth mentioning is the only partial overlap in coding microsatellite mutations, which has been done on a larger sample number (N = 10 cases each; paragraph 2.9 in the enhanced version of the manuscript) here.
Finally, we are aware that these data only partially answer the question and that another set of analyses is unquestionably a very interesting point that should be done; however, these additional analyses are out of the scope of this initial study. We integrated a section “critical limitation of the study” (pages 12/13), but apologize for not providing the information here. Nevertheless, we will be happy to incorporate data in an upcoming manuscript.
We are very confident that the revised version now matches the requirements for publication in Cancers. We would be very pleased if you and the reviewers would find this enhanced version suitable for publication.

Round 2
Reviewer 1 Report
In the revised manuscript, the authors have explained the limit of this study in Discussion, provided new results to analyze the mutations in MSI pathway and vaccine efficacy, and modified figures and text accordingly.However, some concerns need to be addressed.
Major concerns:
In the response cover letter, the authors mentioned the GIT sample #7450 is from a malemouse, however in the text and figures it shows #7450 is from a female It makes me worry about whether the authors have mistaken the gender to make the conclusion of gender-specific differences, because the mutational profile of #328 is similar to that of the two female lymphoma samples, while the mutational profile of #7450 is dramatically different from the other three samples. Please confirm the gender of both the GIT samples. In the legend of Figure 6, the authors claim that there are 8 and 7 mice received A7450 and 328 vaccine respectively, as well as 5 control mice without treatment. However, the survival curves clearly show 7 mice per group. Please carefully check all the data points and correct the figure or the legend.Minor concerns:
According to the text, I think the unit of X-axis of Figure 6 should be “days” but not “week”.Author Response
please see the attachment

Round 3
Reviewer 1 Report
The authors have addressed all of my concerns.